# Implementing a self-monitoring application during pregnancy and postpartum for rural and underserved women: A qualitative needs assessment study

**Marlo M. Vernon**[1]*, **Frances M. Yang**[2]

**1** Cancer Prevention, Control, and Population Health, Georgia Cancer Center, Medical College of Georgia, Augusta University, Augusta, GA, United States of America, **2** School of Nursing, University of Kansas Medical Center, Kansas City, KS, United States of America

* mvernon@augusta.edu

## Abstract

**Data Availability Statement:** Yang, FM. (2022). vidaRPM Qualitative Dataset Version V2) Harvard Dataverse. https://doi.org/10.7910/DVN/ANEET7.

**Funding:** This study was supported through the Health Resources and Services Administration

### Background

Georgia has one of the highest maternal mortality rates within the US. This study describes the qualitative needs assessment undertaken to understand the needs of rural and underserved women and their perspectives on implementing a self monitoring application during pregnancy and postpartum.

### Methods

Qualitative methodology was used to conduct the needs assessment of 12 health care providers (nurses, nurse-midwives, patient care coordinators, and physicians) and 25 women from rural and underserved populations in Georgia was conducted to ascertain common themes on three topics: pregnancy care experiences, comfort with technology, and initial perspectives on the proposed VidaRPM application. Transcription, coding, and consensus were conducted using content analysis and a Cohen's Kappa coefficient was calculated to identify level of overall agreement between raters for the representative quotes identified for each theme.

### Results

The overall agreement for the representative quotes that were chosen for each theme was in strong agreement (κ = 0.832). The major provider feedback included the following regarding the VidaRPM app: inclusion of questions to monitor physical well-being, embedded valid and reliable educational resources, and multiple modalities. The overall feedback from the mothers regarding the VidaRPM application was the virtual aspect helped overcome the barriers to accessing care, comfort with both WiFi and technology, and sustainable utility.

(HRSA), Maternal and Child Health Bureau, 0035251 Remote Pregnancy Monitoring Grand Challenge Phase 1 and 2 Prize. The funder had no involvement in the design, analysis, or writing of this manuscript.

**Competing interests:** The authors have declared that no competing interests exist.

## Discussion

The needs of rural and underserved pregnant women and their providers were assessed to develop and refine the VidaRPM app. This qualitative study on the VidaRPM app is the first step towards closing the gap between providers and patients during prenatal and postpartum periods by empowering and educating women into the first-year postpartum living in rural and underserved areas.

## Introduction

In Georgia, 60% of 2014 pregnancy-related deaths in the state were preventable. The state has one of the highest maternal mortality rates within the US, with 37.2 maternal deaths per 100,000 births in 2016 –a significant trend increasing since 2008 [1]. Current estimates identify an increasing trend to 2018 [2]. The region of focus for this study has a higher maternal mortality rate than state and national averages, 54.6 maternal deaths out of 100,000 live births between 2014 and 2018. Black, non-Hispanic (NHB) women are more than 3.3 times more likely to die from pregnancy-related complications in Georgia, and those who live in rural areas compared to urban areas, are at increased risk of negative maternal health outcomes.

The leading global causes of maternal morbidity and mortality include preeclampsia and eclampsia, affecting up to 200,000 women a year in the US, and accounting for 7.4% of US maternal deaths [3]. Pre-eclampsia is defined by a new onset of elevated blood pressure ($\geq$140 mmHg/90 mmHg on two occasions, four hours apart) and proteinuria after 20 weeks gestation. Pre-eclampsia average rates in Georgia were calculated at 10% among almost 1,400 births in 2018; higher than expected rates between 2–8% [4]. Pre-eclampsia is diagnosed with both high blood pressure ($\geq$140 mmHg/90 mmHg, for systolic and diastolic blood pressure, respectively) and proteinuria and can result in maternal organ dysfunction, premature delivery, fetal growth restriction, compromised placenta and restricted blood flow. Untreated, it can cause eclampsia and seizures and be fatal for both mother and baby. Pre-eclampsia diagnosis puts women at higher risk for cardiovascular disease later in life; this makes intervention a key component of primary and secondary prevention. While birth is the most successful intervention, 75% of deaths due to pre-eclampsia occur in the postpartum period. There is a significant racial/ethnic disparity in pre-eclampsia/eclampsia. Compared to white women, non-Hispanic black women have a 60% increased preeclampsia risk, and are 3xs more likely to die, which remains regardless of socioeconomic status [5]. The targeted rural and underserved populations included in this study often endure extreme stresses of daily living, putting them at additional risk of hypertension disorders.

Several studies have investigated preventive and early diagnostic measures for pre-eclampsia. Plasma biomarker screening in the third trimester has been found to be a cost-effective and feasible predictive measure of later preeclampsia diagnosis; daily low dose aspirin (150mg) use from first trimester through 36 weeks gestation among at-risk patients was associated with significantly lower incidence of preterm preeclampsia than a placebo [6–8]. Remote blood pressure monitoring using m(obile)-Health technologies continues to be explored for its usefulness in this population [9]. Women report preferring home to clinic based monitoring in a prospective feasibility study, but replication in low resource communities (lack of reliable internet access and devices) is challenging [10]. However, these studies are focused on prenatal preeclampsia prediction and intervention; no established primary prevention measures for postpartum pre-eclampsia presentation coupled with ongoing mental health evaluation currently [6, 11, 12].

Remote monitoring during pregnancy has been utilized in gestational diabetes care and has demonstrated improved HBA1c outcomes in women who use remote continuous glucose monitoring [13]. Home blood pressure monitoring was found to reduce prenatal visits in a systematic review, however concerns about clinical heterogeneity of the studies [14]. Remote monitoring of gestational hypertension in Belgium resulted in reduced hospitalizations and NICU admits compared to usual care. A prospective cohort in England of 201 women reported twice daily blood pressures throughout their pregnancy; 66% continued self-monitoring to the end of their pregnancies and of those who developed gestational hypertension or pre-eclampsia, 39% reported an elevated at home blood pressure [15]. Tucker et al, suggest that enhanced support for self-monitoring may improve long term monitoring compliance. An Australian intervention on gestational weight gain reported that 96% of participants replied to the goal and weight check texts; the intervention group reported higher PA and less weight gain [16]. Hanley et al. report that provider workload was reduced with patient at-home monitoring of their blood pressure [17]. A separate study highlighted blood pressure as one of the top two preferences for monitoring among both providers and patients [18].

Over half of the counties in GA do not have a practicing obstetrician/gynecologist (OB/GYN), and in the ECHD this is true for three out of thirteen counties [1]. In the rural counties surveyed, on average the ratio of population to primary care provider was 2,395:1; for minority and underserved women surveyed in urban counties the average was 980:1. This barrier of accessing an OB/GYN doctor requires women to travel long distances to receive prenatal care, and incurs additional burdens such as lost wages, cost of childcare, transportation difficulties, and lost productivity. Consequently, stresses associated with these additional burdens due to lack of access indirectly leads to hypertension (pre-eclampsia and eclampsia), cardiomyopathy, and cardiovascular conditions, which are negative health conditions found to be the highest causes of maternal mortality in the postpartum period [1].

Women in rural and underserved communities experience significant barriers to healthy behaviors and outcomes including those in the social environment (income, education, social connectedness), physical environment (rurality, lack of transportation and lack of access to supports for physical activity, lack of access to healthy food choices), and in the community and health care system environment (significant lack of access to primary care and obstetric providers). Within the state, identified areas of concern include a lack of recognition and assessment of risk factors by both providers and women, inadequate follow-up postpartum, and a lack of referrals to specialists [1]. A separate study highlighted blood pressure as the one of the top two preferences for monitoring among both providers and patients [18].

In order to address issues of access, improve women's self-efficacy, and facilitate provider-patient communication, the VidaRPM (Remote pregnancy and postpartum monitoring) application was conceptualized. Women will monitor their daily blood pressure and weekly weight and mental health through an mHealth interface of simple text messaging and web interface. When women enter results that are above normal thresholds (high blood pressure, rapid weight gain, or evidence of depression or suicidality), women receive a notification to contact their provider for further evaluation. Specific health outcomes to be addressed include reducing incidence of pre-eclampsia, early identification and treatment initiation of gestational hypertension, and early identification of women experiencing mental health concerns in conjunction with health education to improve health literacy and self-efficacy.

Here we describe the qualitative needs assessment undertaken to understand the needs of rural and underserved, minority women and their perspectives on implementing a self-monitoring application during pregnancy and postpartum.

## Methods

### Setting and sample

A qualitative assessment of 12 health care providers (nurses, nurse-midwives, patient care coordinators, and physicians) and 28 women from rural and underserved/minority populations across Georgia was conducted to ascertain common themes on three topics: pregnancy care experiences, comfort with technology, and initial perspectives on the proposed VidaRPM application. (Table 1. Maternal Demographics).

Participants were recruited from a 13 county public health district in east Georgia which is 70.9% rural compared to the state of GA at 24.9%. On average, the population is 38% non-Hispanic Black (Range: 4%-59%). The poverty rate of all counties falls at 24.3%, which is well above the poverty rate for the state of GA at 16.9%, and almost double the national poverty rate of 13.4%. The average income of the district is $38,448 compared to $56,183 for the state of GA, and the average uninsured rate is 16%. Eleven of the counties have been designated as Medically Underserved Areas by HRSA; the other two counties have been designated as having Medically Underserved Populations, meaning that all 13 counties in this health district experience significant lack of access to primary care providers. According to the latest RWJF County Health Rankings, 9 counties rank in the bottom 20% out of 159 GA counties. Only 2/13 counties have full-time access to regular obstetric care.

Healthcare providers were invited to recruit from faculty lists, local and rural OB/GYN providers, and through snowball referral. Women were recruited from rural health care clinics, through referrals from other participants, and from those receiving care in a local public health clinic. Inclusion criteria included adult women ($\geq$ 18 years old), who were currently pregnant or had been in the last five years, lived in rural Georgia counties, or who were from a minority/underserved population; provider inclusion criteria included any adult who provided health care services to pregnant women. Health care clinics ranged in size from single provider to three nurse midwives/OB-GYNs. Average travel time to their OB/GYN provider ranged from 5 minutes to 60 minutes (average 24 minutes).

**Table 1. Maternal demographics (n = 28).**

| | |
|---|---|
| Rural | 17 |
| Non-Hispanic White | 12 |
| Non-Hispanic Black | 15 |
| Hispanic | 1 |
| Age Range | |
| >20 | 5 |
| 21–34 | 18 |
| >35 years | 5 |
| Education | |
| GED/High School | 2 |
| Some College | 21 |
| College Graduate | 4 |
| Graduate/Professional | 1 |
| Maternal Status | |
| First Time Mother | 11 |
| Pregnant | 17 |
| Postpartum | 11 |

### Ethical considerations

This study received IRB approval from Augusta University. Providers and women completed informed consent, and either participated in a focus group or individual interviews.

### Data collection

Sessions were completed in person and over the phone. Women received a thank you gift of mother and baby goods for participating. Interviews lasted approximately 30 minutes, and all were recorded and later transcribed.

### Analysis

Authors completed a multistep analysis for reviewing the transcripts, coding common themes separately for providers and women participants. After individual thematic coding was completed, investigators discussed and agreed upon the main themes among the responses. A Kappa coefficient for agreement was estimated using the SPSS version 27 (IBM, 2020) for the quotes that were chosen across and between the three raters that are presented below to represent the themes.

## Results

The overall agreement for the representative quotes that were chosen for each theme was in strong agreement ($\kappa$ = 0.832) [19]. The Cohen's weighted Kappa coefficient also showed strong agreement between each pair of raters, between Rater 1 and 2 it was 0.884, between Raters 2 and 3 it was 0.942, and between raters 1 and 3 it was 0.829.

### Provider feedback

All providers (n = 12, 50% African American, 86% female; 4 physicians, 6 nurses, 2 nurse midwives) reported enthusiastic support for the proposed application and offered significant suggestions for improving the interface.

**Monitoring physical well-being using VidaRPM application.**   Responses varied on expectation of maternal use–providers report that most women monitor their blood pressure when asked but were concerned about misreporting. Inclusion of questions to monitor physical well-being was the most suggested addition to the proposed application. Providers also expressed that while not all women had access to smart phones, all of their patients texted regularly. Some providers reported using personal text messaging as a fast and easy way to stay in touch with their patients, circumventing phone calls and "phone tag."

**Health education resources.**   Valid and reliable educational resources were the second most requested addition. One provider emphasized the need for education within the app by stating, "I think that with the message coming back to them and saying, okay, this is abnormal, there needs to be some explanation because the first thing that they'll do is probably Google whatever that message says. So I think there needs to be something or click here for more information about high blood pressure and pregnancy, so that they get validated information".

Another provider said it would be imperative to include educational information in the app stating:

> ". . .helping them to understand what some of the warning signs are, what they can do about them, and things like that is important. The provider also followed up by stating "there's a lot of misinformation oftentimes, and them having a more private way to access more accurate information is important."

**Suggestions for improvement.** Another suggestion for the app was providing multiple modalities for patients to access. This suggestion was made by both providers and patients. Some providers emphasized the importance of allowing the patients to have the ability to read the information, hear the information and see a visual. Multiple modalities will increase understanding among patients with different learning styles. One provider explained this by saying "it kind of demonstrates to them what they're supposed to be doing and then kind of gives them step-by-step instructions and maybe has a help section where they can go and review the instructions. So, if they're going to take their blood pressure, or they're going to record their weight, those are things that I think we have to make sure that the instructions are clear to everyone, whether they're visual, auditory. . ."

## Women's feedback

Women (n = 28, 54% African American, 75% completed some college, mean Age = 27.2 (18– 45, 68% rural), reported strong support for the proposed application, and highlighted a need for reliable health education resources and trustworthy sources.

**Accessing prenatal care.** Women described barriers to accessing care including long travel times to prenatal appointments, difficulty accessing childcare during appointments, and taking time off of work. All reported accessing health care within recommended time frames (by the end of the first trimester), did not report complicated pregnancies but often knew someone who had complications.

One woman expressed interests in the virtual aspects of the app by stating "in this day and age, everything is virtual. Interviews, just everything. And with a newborn I can only imagine that transporting a baby from home to the hospital, and from the hospital back home is going to be a lot, and you're still healing. So, to be able to get that care from the comfort of your home, yes that is something I would be interested in".

Another woman was intrigued with the app because it could allow women to know if they are at risk for health complication based on their weight and blood pressure. She knew that she was at risk for high blood pressure, however she was not familiar with preeclampsia or that she may find herself susceptible to it. When asked what she liked most about the idea of the app she responded "[d]ying of preeclampsia? Like I didn't know that until you educated me on that. So of course I would want to check on me, because without me, my baby. . . Who they got? I'm all my baby got".

**Comfort with technology.** The majority of the women, which was 95% of the interviewees, expressed comfort with both WiFi and technology in general. All women reported texting as a common form of communication. Women in rural areas reported difficulties with reliable internet access. Women reported being unsure if online information was reliable and were hesitant to trust the health messages they found. They were enthusiastic to have a resource designed that they knew was trustworthy. One woman stated ". . .when you Google stuff on the internet it always leads to a rabbit hole of you dying so I just try not to".

**Utility of the VidaRPM application.** Women were also very supportive about the proposed application and were most excited about the idea of ongoing monitoring during pregnancy. Most women expressed a desire for a mobile application with additional education resources.

One woman said "I would like my providers to see my daily readings because I wouldn't know what to do. They know what to do. A lot of times, we don't even know anything's wrong with us." While also adding that she would call as soon as she was notified that her results were abnormal. Another woman eagerly proclaimed "I think that [app] is comforting because I'm like number seven of pregnant women currently in my family. We talk a lot because every

pregnancy is different, so we give each other feedback. And what I've noticed is most of my family members will say they have this going on, or they'll have that going on, and they're embarrassed to go to the doctor. Or maybe my blood pressure is high, I don't know, I'm going to lay down and see how I feel later. So, I think if you had an opportunity to check that stuff at home a lot of things that could be prevented would be. So, I like that".

**Suggestions for improvement.**  Suggestions included incentives for ongoing use, mother/baby activity calendars, a place to keep baby records, and valid and reliable education. These points were used to develop the mockups for the mobile application and the tabs for the website for VidaRPM.

## Conclusions for practice

In this qualitative formative, women and providers consistently identified a need for reliable and valid health resources. Women were unsure if internet resources were trustworthy, indicating a need for health literacy education. Women reported significant barriers to accessing care–including long travel, a lack of providers, and lost productivity. Providers were frustrated with the lack of tools to interact and monitor their patients outside of traditional clinical care.

Using the feedback from providers and women, VidaRPM design and edits were made to close communication and access gaps between providers and patients during prenatal and postpartum periods, Based on the mother and providers responses, a simple SMS interface and web application for women to record blood pressure, weight, and respond to mental health questions, and receive feedback for responses outside of normal ranges was further adapted to meet identified needs. The SMS flow now includes follow-up questions based on physical signs and symptoms of pre-eclampsia, prompts to take blood pressure medication if prescribed, and the option for providers to set lower or higher thresholds for patients, as requested by providers. Providers receive either an email or text alert or both when a patient records an abnormal reading. Patients receive a notification to contact their health provider under the same algorithms. A secure password protected website was developed for provider and patient interaction, to improve real time communication. Women can log into the website to update their daily responses if they are unable to access their SMS interface, to alleviate any barriers to technology.

In this study, the providers and pregnant women identified the need for health monitoring and to access valid, reliable, and pertinent health information in rural and underserved areas. Because mothers highlighted a need for health education they knew was reliable, a four module quick learn was adapted for mothers to include education about high blood pressure during pregnancy, how to take a blood pressure, what to do if a high blood pressure is recorded, and postpartum health and monitoring. The quick learn can be accessed by participants from the application. We utilized evidence-based, valid, and culturally competent resources for prenatal and postpartum health education and provide targeted delivery. This is a simple and innovative telehealth monitoring and education tool intended to be scalable and easily disseminated to a much wider population.

The barriers identified in this study overlap with the prenatal healthcare experience in a qualitative research study found among 54 minority women in Northern California, which include unmet information needs, inconsistent social support, disrespect, racism, or discrimination experienced from providers and staff [20]. Women report a preference for a mobile app to provide health education, however, previous implementation of "The Health-e Babies App" for antenatal education discovered that significant barriers to feasibility and accessibility exist in low income populations [21]. This provides support for low technology resource designs, such as VidaRPM.

This proposed mHealth monitoring and online education resources may increase self-efficacy, health literacy, and perceived control over women's health. By providing provider and monitoring access through mHealth, women may perceive improved quality of care and become a patient-partner; it is hoped that this will significantly impact their own present and future health outcomes, and improve the health of their families and children. Future studies will evaluate the long-term health impact among families and children.

## Supporting information

**S1 File.**
(DOCX)

**S2 File.**
(DOCX)

## Acknowledgments

We thank all of our participants for their time and insight.

## Author Contributions

**Conceptualization:** Marlo M. Vernon.

**Data curation:** Marlo M. Vernon.

**Formal analysis:** Marlo M. Vernon, Frances M. Yang.

**Funding acquisition:** Marlo M. Vernon.

**Investigation:** Marlo M. Vernon, Frances M. Yang.

**Methodology:** Marlo M. Vernon, Frances M. Yang.

**Project administration:** Marlo M. Vernon.

**Resources:** Marlo M. Vernon.

**Software:** Marlo M. Vernon, Frances M. Yang.

**Supervision:** Marlo M. Vernon.

**Validation:** Marlo M. Vernon, Frances M. Yang.

**Visualization:** Marlo M. Vernon.

**Writing – original draft:** Marlo M. Vernon, Frances M. Yang.

**Writing – review & editing:** Marlo M. Vernon, Frances M. Yang.

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
