## [Decision Letter · Decision Letter 0]

14 Mar 2022

PONE-D-21-23332Implementing a self-monitoring application during pregnancy and postpartum for rural and underserved women: A qualitative needs assessment studyPLOS ONE

Dear Dr. Yang,

Thank you for submitting your manuscript to PLOS ONE. After careful consideration, we feel that it has merit but does not fully meet PLOS ONE’s publication criteria as it currently stands. Therefore, we invite you to submit a revised version of the manuscript that addresses the points raised during the review process.

We look forward to receiving your revised manuscript.

Kind regards,

Chi-Hua Chen, Ph.D.

Academic Editor

PLOS ONE

Journal Requirements:

2. Please include a copy of the interview guide used in the study, in both the original language and English, as Supporting Information, or include a citation if it has been published previously.

This study was supported through a HRSA, Bureau of Maternal and Child Health, Remote Pregnancy Monitoring Grand Challenge Phase 1 and 2 Prize. The funder had no involvement in the design, analysis, or writing of this manuscript.

No

5. Please note that in order to use the direct billing option the corresponding author must be affiliated with the chosen institute. Please either amend your manuscript to change the affiliation or corresponding author, or email us at plosone@plos.org with a request to remove this option.

7. Please amend your manuscript to include your abstract after the title page.

Reviewers' comments:

Reviewer's Responses to Questions

**Comments to the Author**

1. Is the manuscript technically sound, and do the data support the conclusions?

Reviewer #1: Partly

2. Has the statistical analysis been performed appropriately and rigorously? 

Reviewer #1: Yes

3. Have the authors made all data underlying the findings in their manuscript fully available?

Reviewer #1: No

4. Is the manuscript presented in an intelligible fashion and written in standard English?

Reviewer #1: Yes

5. Review Comments to the Author

Reviewer #1: In the introduction, the writers make a case for the disparities in health outcomes between white women and Black and Hispanic women. They end the section by introducing the targeted rural women with no mention of what race this target demographic is. The write must consider including health statistics for the target population if these are readily available.

Barriers to healthy behaviors and outcomes are listed in the background and significance. I believe this paper would be strengthened with a presentation of the analysis based on these identified barriers to show how the proposed intervention will address the barriers if at all. Presentation of demographics tables for respondent types (FGD and individual respondents) is missing

The writers should consider including a brief description of the health system in the setting and sample section. Average distances to health facilities, population size served by health facility, average number of staff at the rural health facilities.

They writers have not stated clearly their inclusion criteria for the women included for this formative work. They say women were recruited from the rural health care. What was the minimum age for inclusion, were these pregnant women, women who were previously pregnant, or was this not relevant for selection?

For the discussion, the writers should consider focusing circling back to the barriers identified earlier in the paper and discuss how their proposed work will fill the gaps (some or all)

6. PLOS authors have the option to publish the peer review history of their article (what does this mean?). If published, this will include your full peer review and any attached files.

Reviewer #1: No

---

## [Author Response · Author response to Decision Letter 0]

9 May 2022

April 8, 2022

PLOS ONE 

RE: PONE-D-21-23332

Implementing a self-monitoring application during pregnancy and postpartum for rural and underserved women: A qualitative needs assessment study

To Editor Chen: 

Thank you for considering this manuscript to have merit. Please find enclosed the following as requested: 

• A rebuttal letter that responds to each point raised by the academic editor and reviewer uploaded as a separate file labeled 'Response to Reviewers'.

• A marked-up copy of the manuscript that highlights changes made to the original version, as a separate file labeled 'Revised Manuscript with Track Changes'.

• An unmarked version of your revised paper without tracked changes uploaded as a separate file labeled 'Manuscript'.

The corresponding author who will be remitting payment is Dr. Marlo Vernon, but she is currently on maternity leave, so I will be the contact author until she returns. 

The additional issues to address are detailed below per instruction to include in the cover letter:

Response #1: All PLOS One’s style requirements have been reviewed and included for both the manuscript and the title page uploaded.

2. Please include a copy of the interview guide used in the study, in both the original language and English, as Supporting Information, or include a citation if it has been published previously.

Response #2: The interview guides for both the participant and providers have been uploaded.

Response #3: Please help us locate the "Financial Disclosure" section within the PLOS ONE portal, we have included the information in the "Funding Information" section of the PLOS ONE portal. We have provided the information for both Financial Disclosure and Funding Information on the title page. 

This study was supported through a HRSA, Bureau of Maternal and Child Health, Remote Pregnancy Monitoring Grand Challenge Phase 1 and 2 Prize. The funder had no involvement in the design, analysis, or writing of this manuscript.

No

Response #4: This study was supported through a HRSA, Bureau of Maternal and Child Health, Remote Pregnancy Monitoring Grand Challenge Phase 1 and 2 Prize. The funder had no involvement in the design, analysis, or writing of this manuscript.

5. Please note that in order to use the direct billing option the corresponding author must be affiliated with the chosen institute. Please either amend your manuscript to change the affiliation or corresponding author, or email us at plosone@plos.org with a request to remove this option.

Response #5: Corresponding author is Dr. Marlo Vernon who will remit payment.

Response #6: Yang, FM. (2022). vidaRPM Qualitative Dataset Version V2) Harvard Dataverse. https://doi.org/doi:10.7910/DVN/ANEET7

7. Please amend your manuscript to include your abstract after the title page. 

Response #7: Amended manuscript includes the abstract after the title page. 

Reviewer #1: In the introduction, the writers make a case for the disparities in health outcomes between white women and Black and Hispanic women. 

1. They end the section by introducing the targeted rural women with no mention of what race this target demographic is. The write must consider including health statistics for the target population if these are readily available.

Response #1: We thank the Reviewer for this insightful review. On page 6, in the Methods section, Setting and sample subsection, the statistics for rural populations in Georgia have been added regarding lack of access to primary care providers and rates of chronic disease. 

“Participants were recruited from a 13-county public health district in east Georgia which is 70.9% rural compared to the state of GA at 24.9%. On average, the population is 38% non-Hispanic Black (Range: 4%-59%). The poverty rate of all counties falls at 24.3%, which is well above the poverty rate for the state of GA at 16.9%, and almost double the national poverty rate of 13.4%. The average income of the district is $38,448 compared to $56,183 for the state of GA, and the average uninsured rate is 16%. Eleven of the counties have been designated as Medically Underserved Areas (MUA) by the Health Resource Services Administration (HRSA); the other two counties have been designated as having Medically Underserved Populations (MUP), meaning that all 13 counties in this health district experience significant lack of access to primary care providers. According to the latest Robert Wood Johnson Foundation (RWJF) County Health Rankings, nine counties ranked in the bottom 20% of a total of 159 GA counties. Only two out of 13 counties have full-time access to regular obstetric care.”

2. Barriers to healthy behaviors and outcomes are listed in the background and significance. I believe this paper would be strengthened with a presentation of the analysis based on these identified barriers to show how the proposed intervention will address the barriers if at all. Presentation of demographics tables for respondent types (FGD and individual respondents) is missing.

Response #2: On page 5, in the Methods section, Setting and sample subsection, the Maternal Demographic Characteristics is the title included for Table 1 is now clearly indicated. How the application was designed in response to barriers is included in the discussion section. Provider input resulted in additional follow-up questions added to the application monitoring, and patient health education needs were also met through a quick learn adapted for mothers. Gaps in access to providers and communication between patients/providers are also addressed. We appreciate the Reviewer’s helpful suggestion with regards to including this important information.

Table 1. Maternal Demographic Characteristics (n=28)

Rural 17 61%

Non-Hispanic White 12 43%

Non-Hispanic Black 15 54%

Hispanic 1 4%

Age Range 

>20 5 18%

21-34 18 64%

>35 years 5 18%

Education 

GED/High School 2 7%

Some College 21 75%

College Graduate 4 14%

Graduate/Professional 1 4%

Maternal Status 

First Time Mother 11 39%

Pregnant 17 61%

Postpartum 11 39%

3. The writers should consider including a brief description of the health system in the setting and sample section. Average distances to health facilities, population size served by health facility, average number of staff at the rural health facilities.

Response #3: Participants came from more than one health system from 13 counites in Georgia. This information is now clarified, thanks to the Reviewer’s point, on page 5 of the Methods section in the Setting and Sample subsection. In addition, information is now included about the number of providers/site and details about the ratio of primary care provider to population were included for the counties surveyed. The average travel time was used as a proxy for distance from provider is found on page 7, in the inclusion criteria paragraph of the Methods section in the Setting and Sample subsection: “Health care clinics ranged in size from single provider to three nurse midwives/OB-GYNs. Average travel time to their OB/GYN provider was 24 minutes with a range from five minutes to one hour.”

On page 3, paragraph 2, of the Introduction section: “In the rural counties surveyed, the average ratio of persons to primary care provider was 2,395:1. Specifically, for minority and underserved women surveyed in urban counties the average was 980:1.”

On page 5 of the Methods section in the Setting and Sample subsection: “Participants were recruited from a 13-county public health district in east Georgia which is 70.9% rural compared to the state of GA at 24.9%. On average, the population is 38% non-Hispanic Black (Range: 4%-59%). The poverty rate of all counties falls at 24.3%, which is well above the poverty rate for the state of GA at 16.9%, and almost double the national poverty rate of 13.4%. The average income of the district is $38,448 compared to $56,183 for the state of GA, and the average uninsured rate is 16%. Eleven of the counties have been designated as Medically Underserved Areas (MUA) by the Health Resource Services Administration (HRSA); the other two counties have been designated as having Medically Underserved Populations (MUP), meaning that all 13 counties in this health district experience significant lack of access to primary care providers. According to the latest RWJF County Health Rankings, 9 counties rank in the bottom 20% out of 159 GA counties. Only 2 out of 13 counties have full-time access to regular obstetric care.”

4. They writers have not stated clearly their inclusion criteria for the women included for this formative work. They say women were recruited from the rural health care. What was the minimum age for inclusion, were these pregnant women, women who were previously pregnant, or was this not relevant for selection?

Response #4: We again thank the Reviewer for this critique, on page 7, in the inclusion criteria paragraph of the Methods section in the Setting and Sample subsection, reads as follows: “Inclusion criteria included adult women (aged 18 years old and older), who were currently pregnant or had been in the last five years, lived in rural Georgia counties, or who were from a minority/underserved population; provider inclusion criteria included any adult who provided health care services to pregnant women. Health care clinics ranged in size from single provider to three nurse midwives/OB-GYNs. Average travel time to their OB/GYN provider was 24 minutes with a range from five minutes to one hour.”

5. For the discussion, the writers should consider focusing circling back to the barriers identified earlier in the paper and discuss how their proposed work will fill the gaps (some or all)

Response #5: The Reviewer’s suggestion has been integrated on pages 12-14 to highlight improved communication, access to providers and monitoring through mHealth, and self-efficacy and knowledge through health education resources: “Using the feedback from providers and women, VidaRPM design and edits were made to close communication and access gaps between providers and patients during prenatal and postpartum periods, Based on the mother and providers responses, a simple SMS interface and web application for women to record blood pressure, weight, and respond to mental health questions, and receive feedback for responses outside of normal ranges was further adapted to meet identified needs. The SMS flow now includes follow-up questions based on physical signs and symptoms of pre-eclampsia, prompts to take blood pressure medication if prescribed, and the option for providers to set lower or higher thresholds for patients, as requested by providers. Providers receive either an email or text alert or both when a patient records an abnormal reading. Patients receive a notification to contact their health provider under the same algorithms. A secure password protected website was developed for provider and patient interaction, to improve real time communication. Women can log into the website to update their daily responses if they are unable to access their SMS interface, to alleviate any barriers to technology. 

In this study, the providers and pregnant women identified the need for health monitoring and to access valid, reliable, and pertinent health information in rural and underserved areas. Because mothers highlighted a need for health education they knew was reliable, a four module quick learn was adapted for mothers to include education about high blood pressure during pregnancy, how to take a blood pressure, what to do if a high blood pressure is recorded, and postpartum health and monitoring. The quick learn can be accessed by participants from the application. We utilized evidence-based, valid, and culturally competent resources for prenatal and postpartum health education and provide targeted delivery. This is a simple and innovative telehealth monitoring and education tool intended to be scalable and easily disseminated to a much wider population.

The barriers identified in this study overlap with the prenatal healthcare experience in a qualitative research study found among 54 minority women in Northern California, which include unmet information needs, inconsistent social support, disrespect, racism, or discrimination experienced from providers and staff.(McLemore et al., 2018) Women report a preference for a mobile app to provide health education, however, previous implementation of “The Health-e Babies App” for antenatal education discovered that significant barriers to feasibility and accessibility exist in low income populations.(Dalton et al., 2018) This provides support for low technology resource designs, such as VidaRPM.

This proposed mHealth monitoring and online education resources may increase self-efficacy, health literacy, and perceived control over women’s health.By providing provider and monitoring access through mHealth, women may perceive improved quality of care and become a patient-partner; it is hoped that this will significantly impact their own present and future health outcomes, and improve the health of their families and children. Future studies will evaluate the long-term health impact among families and children.”

Please do not hesitate to contact me if further information is needed.

Sincerely,

Frances M. Yang, PhD

Research Associate Professor

---

## [Decision Letter · Decision Letter 1]

7 Jun 2022

Implementing a self-monitoring application during pregnancy and postpartum for rural and underserved women: A qualitative needs assessment study

PONE-D-21-23332R1

Dear Dr. Yang,

We’re pleased to inform you that your manuscript has been judged scientifically suitable for publication and will be formally accepted for publication once it meets all outstanding technical requirements.

Kind regards,

Chi-Hua Chen, Ph.D.

Academic Editor

PLOS ONE

Additional Editor Comments (optional):

Reviewers' comments:

Reviewer's Responses to Questions

**Comments to the Author**

1. If the authors have adequately addressed your comments raised in a previous round of review and you feel that this manuscript is now acceptable for publication, you may indicate that here to bypass the “Comments to the Author” section, enter your conflict of interest statement in the “Confidential to Editor” section, and submit your "Accept" recommendation.

Reviewer #1: All comments have been addressed

2. Is the manuscript technically sound, and do the data support the conclusions?

Reviewer #1: Yes

3. Has the statistical analysis been performed appropriately and rigorously? 

Reviewer #1: Yes

4. Have the authors made all data underlying the findings in their manuscript fully available?

Reviewer #1: Yes

5. Is the manuscript presented in an intelligible fashion and written in standard English?

Reviewer #1: Yes

6. Review Comments to the Author

Reviewer #1: The writers addressed the comments from my first review of the manuscript. This included clearly indicating the target group for the study, the inclusion criteria for the study, the study setting and making revision to the discussion. The manuscript as it reads now can be considered for publication by the editor.

7. PLOS authors have the option to publish the peer review history of their article (what does this mean?). If published, this will include your full peer review and any attached files.

Reviewer #1: No

---

## [Editor Report · Acceptance letter]

8 Jul 2022

PONE-D-21-23332R1 

Implementing a self-monitoring application during pregnancy and postpartum for rural and underserved women: A qualitative needs assessment study 

Dear Dr. Yang:

I'm pleased to inform you that your manuscript has been deemed suitable for publication in PLOS ONE. Congratulations! Your manuscript is now with our production department. 

Kind regards, 

on behalf of

Professor Chi-Hua Chen 

Academic Editor

PLOS ONE